# Very Low Birth Weight Outcomes and Admission Temperature: Does Hyperthermia Matter?

**DOI:** 10.3390/children9111706

**Published:** 2022-11-07

**Authors:** Hannah Brophy, Gaik Min Tan, Charles William Yoxall

**Affiliations:** Neonatal Unit, Liverpool Womens Hospital, Liverpool L8 7SS, UK

**Keywords:** hyperthermia, prematurity, resuscitation

## Abstract

National and international recommendations for thermal care at preterm birth include recommendations to avoid hypothermia and hyperthermia. There is limited evidence demonstrating harm resulting from admission hyperthermia. Our aim was to assess the relationships between admission temperature and outcomes in very low birth weight (VLBW) babies in a unit with low rates of hypothermia and a higher rate of hyperthermia. This was an observational study based on routinely collected data including demographics, admission temperature, survival and major morbidity outcomes. Subjects were 1104 consecutive inborn VLBW babies admitted to a Neonatal Intensive Care Unit in United Kingdom between 2010 and 2017. Results: 155 (14%) of babies were hypothermic (<36.5 °C) with only 21 (1.9%) < 36 °C, and 254 (23%) of babies were hyperthermic (>37.5 °C). The rate of major abnormality on cranial ultrasound scan was increased in the hyperthermic babies compared to the normothermic babies (37/239 (15.5%) vs. 54/601 (9%), relative risk (95% CI) 1.723 (1.166 to 2.546), *p* = 0.006). There was no difference in survival or other major morbidity in the hyperthermic babies compared to the normothermic babies. There was no association between hypothermia and survival or any major morbidity, although this probably reflects the low power of the study given the low rates of significant hypothermia. Higher admission temperature was associated with an increase in the risk of major cranial ultrasound abnormality using multiple logistic regression analysis (*p* = 0.007) with an increased odds ratio (95% CI) of 1.48 (1.11 to 1.97) for each degree of increase. We conclude that admission hyperthermia is independently associated with an increased risk of preterm brain injury. It is not possible to state whether this is a causative association, or whether the association is a consequence of a shared aetiology of perinatal infection.

## 1. Introduction

It is well established that hypothermia on admission to the neonatal unit is an independent predictor of mortality in preterm babies. This association continues to be described in the recent literature from many countries including UK [1], Europe [2], North America [3,4] and India [5]. Recommendations for optimal admission temperature at preterm birth include avoidance of both hypothermia and hyperthermia. The UK National Neonatal Audit Programme (NNAP) defines optimal admission temperature in babies born before 32 weeks’ gestation as between 36.5 °C and 37.5 °C [6]. The latest International Liaison Committee on Resuscitation (ILCOR) guidance includes treatment recommendations to prevent hypothermia (defined as <36 °C) and to avoid admission temperatures greater than 38 °C [7]. There is, however, no published evidence of a causal relationship between iatrogenic admission hyperthermia and adverse neonatal outcome.

In our unit, we have achieved a low rate of admission hypothermia in preterm babies over recent years, but have seen a higher rate of hyperthermia as a consequence. Our aim was to investigate whether the admission temperatures for inborn very low birth weight (VLBW) babies admitted to our neonatal intensive care unit (NICU) between 2010 and 2017 predicted mortality or major morbidity before discharge.

## 2. Materials and Methods

Data are routinely collected prospectively on all VLBW babies admitted to our unit for the purposes of benchmarking. For the purposes of this study we collected patient demographics, NICU admission temperature, survival to discharge and major morbidity data for all babies in the study period from between January 2010 and December 2017 inclusive. All inborn babies admitted to the NICU with a birth weight below 1500 g were included. Morbidity definitions included: the presence of a major abnormality on cranial ultrasound (defined as grade III or IV periventricular haemorrhage or the presence of periventricular leukomalacia), bronchopulmonary dysplasia (BPD) (defined as oxygen requirement at 36 weeks corrected gestational age), necrotising enterocolitis (NEC) and sight threatening Retinopathy of Prematurity (ROP) (defined as stage 3 or higher).

Initial analysis was performed in order to explore differences in outcomes by the temperature groups defined by NNAP. Babies were classified into groups as either hypothermic (<36.5°C), normothermic (36.5 °C to 37.5 °C) or hyperthermic (>37.5 °C) based on admission temperature. Clinical outcomes of the hypothermic and hyperthermic groups were compared with the outcomes of the normothermic groups using Chi-squared with Yates’ correction and calculation of relative risk compared to the rates in the normothermic group.

Analyses were also performed with admission temperature as a continuous variable using regression analysis for other continuous variables or Mann–Whitney test for dichotomous variables as appropriate. Significant associations observed on univariate testing were entered into multiple regression analyses to determine which were independent associations.

All analysis was performed by using SPSS v26 (IBM, Portsmouth, UK).

This study was approved as a service evaluation by the Effectiveness Senate of our institution in May 2021 (approval number QIP 0058).

## 3. Results

Data were available for 1104 babies. Demographics and admission temperatures are shown in Table 1. Median (range) gestation was 28 (22 to 37) weeks. Median (range) birth weight was and 1090 (370 to 1495) g. 632 (57%) were delivered by caesarean section, 731 (66%) were singletons and 502 (45.5%) were female.

A total of 695 (63%) babies were normothermic on admission; 155 (14%) were hypothermic, with 21 (1.9%) below 36 °C and only 1 baby below 35 °C; 254 (23%) were hyperthermic with 121 (11%) being above 38 °C.

Clinical outcomes in the hypothermic and hyperthermic groups were compared to the outcomes of the normothermic group (Table 2). The only statistically significant association was the increased rate of major cranial ultrasound abnormality seen in the hyperthermic babies (RR (95% CI) = 1.723 (1.166 to 2.546), *p* = 0.006). There was no evidence of an increased risk of death in the hypothermic group, although this is likely to be a type 2 error due to lack of study power because of the low rate of hypothermia <36 °C in our cohort.

The relationship between admission temperature and clinical outcomes is shown in Table 3.

There were no statistically significant relationships between admission temperature, analysed as a continuous variable, and either death or NEC.

On univariate testing, an increased risk of BPD was associated with higher admission temperature (*p* = 0.001) as well as lower birth weight (*p* < 0.001), shorter gestation (<0.001) and male gender (*p* < 0.001). Only birth weight, gestation and gender were found to be independently associated during multiple logistic regression analysis.

Similarly, on univariate testing an increased rate of sight-threatening ROP was associated with higher admission temperature (*p* = 0.028) as well as lower birth weight (*p* < 0.001), shorter gestation (*p* < 0.001) and lower Apgar score at 5 min (*p* = 0.047). Only birth weight, gestation and Apgar score at 5 min were found to be independently associated during multiple logistic regression analysis.

On univariate testing, an increased risk of major cranial ultrasound abnormality was associated with higher admission temperature (*p* = 0.001) as well as lower birth weight (*p* < 0.0001), shorter gestation (*p* < 0.0001) and lower Apgar score at 5 min (*p* = 0.0003). Higher admission temperature remained independently associated with an increased risk of major cranial ultrasound abnormality during multiple logistic regression analysis (*p* = 0.007) as well as the associations with shorter gestation (*p* < 0.001) and lower Apgar score at 5 min (*p* = 0.007). For each degree of increase in admission temperature, the likelihood of an abnormal cranial ultrasound scan increased by an odds ratio (95% CI) of 1.48 (1.11 to 1.97) or 48%.

## 4. Discussion

Preterm babies who are hypothermic on admission to the neonatal unit have a higher risk of death [1,2,3,4,5]. The association is stronger with increasingly low temperature [3,4] and is most marked in those babies with a temperature below 35 °C [1,2,3,4]. The strength of the association between hypothermia and death, the increase in risk with lower temperatures and biological plausibility strongly suggest that there is a causal relationship between admission hypothermia and decreased survival chances. Admission hypothermia, and its associated increased mortality risk, is avoidable by good thermal care during initial stabilisation at birth.

We did not find any increase in mortality or neonatal morbidity in association with hypothermia in our study. This is likely to be because our study lacks the power to find such a relationship due to the very low rates of hypothermia in our population (14% < 36.5 °C and only 1/1104 below 35 °C). The rate of hypothermia in our study population is much lower than that described in other recent studies, in which hypothermia rates of 40% to 50% are reported [1,2,3,4,5,8,9]. We believe that this is due to a concerted effort in our unit during this period to prevent hypothermia in our patients, including:(a)The use of polythene bags in babies born before 30 weeks’ gestation since 2005. This practice was adopted based on a randomized controlled trial reported by Vohra et al., which demonstrated a reduced incidence of hypothermia [10]. We have previously described the impact of implementing this intervention in our population [11].(b)The use of an exothermic gel mattress (Transwarmers, Drager UK) in babies born before 28 weeks’ gestation since 2007. A recent meta-analysis has described the effectiveness of this approach [12] and we have previously described the impact of implementing this intervention into our practice [13].

The combined effect of these practices appears to have resulted in the higher rate of admission hyperthermia in our patients (23%) than the rates of less than 4% described in the recently reported studies [2,5,8,9].

It is likely that as strategies to reduce admission hypothermia are implemented into practice, there will be an increase in admission hyperthermia. For example: normothermia on admission at preterm birth has been a standard adopted across England and Wales as part of the NNAP since its inception. This has resulted in a 47% relative reduction in the rate of admission hypothermia (from 28% in 2015 to 14.7% in 2020). This has, however, been accompanied by a 13% relative increase in the rate of admission hyperthermia (from 10% in 2015 to 11.3% in 2020) [14,15].

When a baby is found to be hyperthermic on admission, it could be due either to iatrogenic overheating during the period of stabilisation and transfer to NICU or due to the prevention of cooling during that period in a baby who is already pyrexial. Fetal intrapartum temperature is higher than maternal temperature by between 0.1 °C and 3.2 °C [16], so it is likely that a significant number of preterm babies are born with a temperature > 37.5 °C, particularly those in whom maternal temperature is elevated during the intrapartum period. In the study of Sharma et al., all babies admitted with a temperature >37.5 °C were born to women with intrapartum pyrexia [5]. Data from our unit [17] suggest that most babies who are admitted to NNU with hyperthermia are hyperthermic on the labour ward, before transfer to NICU. On the basis of this, we believe it is likely that the high rates of admission hyperthermia seen in our cohort are likely to be a consequence of the prevention of cooling of already pyrexial babies by the thermal care interventions used, rather than iatrogenic warming. We are not aware of any published data to support this hypothesis from other centres and believe it is a subject worthy of further study.

There is very little published evidence relating to neonatal outcomes and admission hyperthermia at preterm birth. The most recent ILCOR evidence review states “Hyperthermia and its outcome remain less well studied with limited new evidence”. The current ILCOR recommendation is “We suggest that hyperthermia (greater than 38.0 °C) be avoided due to the potential associated risks (weak recommendation, very-low-quality evidence)” [7].

Lyu et al. found an increased rate of a composite adverse outcome (severe neurological injury, severe ROP, NEC, BPD and nosocomial infection) in babies with an admission temperature above 38 °C [3]. Sharma et al. also reported an increase in a similar composite outcome in babies with admission temperature >37.5 °C [5]. The Clinical Risk Indicator for Babies II score (CRIB II) [18] increases with an admission temperature >37 °C. We assume that this is because the dataset on which the score was developed found an increased risk of death in those babies, although this is not explicitly stated in the paper describing the development of the score.

We have found in this study that babies with hyperthermia at admission have an increased rate of major cranial ultrasound abnormality. Whilst it is possible that this reflects a causal association between admission hyperthermia and brain injury, it does not of course prove that there is a causal association.

Chorioamnionitis, the most common cause of intrapartum pyrexia, is strongly associated with preterm brain injury. There are numerous studies that demonstrate a link between the presence of periventricular leukomalacia and perinatal infection. These studies and potential mechanisms of injury have been described by Strunk et al. [19]. A meta-analysis by Wu and Colford found that in preterm babies, histologic chorioamnionitis was significantly associated with PVL (RR, 2.1; 95% CI, 1.5–2.9) [20]. Periventricular haemorrhage is also strongly associated with chorioamnionitis. This is illustrated in a recent systematic review and meta-analysis from Villamor-Martinez et al., which demonstrated an increased risk of periventricular haemorrhage in preterm babies born with a history of chorioamnionitis with an odds ratio (95% CI) of 1.62 (1.42–1.85) [21].

## 5. Conclusions

Maternal pyrexia will lead to hyperthermia at birth and this will result in admission hyperthermia in units that have good systems in place to prevent cooling during stabilisation and transfer. As units improve thermal care to reduce the rate of hypothermia, it is likely that they will see an increase in the rate of hyperthermia, which is likely to be a consequence of reduced cooling of already pyrexial babies rather than iatrogenic overheating.

We have demonstrated an association between admission hyperthermia in VLBW babies and adverse cranial ultrasound outcome. We think it is probable that the association we have seen between admission hyperthermia and brain injury results from a shared aetiology of maternal infection rather than there being a direct causal relationship between the two, although further work is required to allow us to understand this association better

Current recommendations relating to thermal care at preterm birth advise the avoidance of hyperthermia as well as the prevention of hypothermia. We believe that, on the basis of current knowledge, the avoidance of hypothermia should be the priority.

## Figures and Tables

**Table 1 children-09-01706-t001:** Demographics and admission temperatures of study cohort.

Variable	
Median (range) Gestation	28 (22 to 37) weeks
Median (range) Birth Weight	1090 (370 to 1495) g
Weight below 1000 g	459 (45.7%)
Caesarean section	632 (57%)
Singleton	731 (66%)
Female gender	502 (45.5%)
Admission Temperatures	
<36 °C	21 (1.9%)
36 °C to 36.4 °C	134 (12.1%)
36.5 °C to 37.5 °C	695 (63%)
>37.5 °C	254 (23%)

**Table 2 children-09-01706-t002:** Clinical outcomes by temperature group.

	Normothermia	Hypothermia			Hyperthermia		
			RR (95% CI)	p		RR (95% CI)	p
Death	126/695 (18.1%)	28/155 (18.1%)	0.996 (0.688 to 1.443)	ns	42/254 (16.5%)	0.912 (0.663 to 1.253)	ns
Major USS abnormality	54/601 (9%)	11/129 (8.5%)	0.949 (0.511 to 1.764)	ns	37/239 (15.5%)	1.723 (1.166 to 2.546)	0.006
BPD	251/591 (42.4%)	44/128 (34.5%)	0.809 (0.626 to 1.047)	ns	101/215 (47%)	1.101 (0.933 to 1.311)	ns
ROP	28/695 (4%)	7/155 (4.5%)	1.072 (0.477 to 2.412)	ns	16/245 (6.3%)	1.621 (0.892 to 2.944)	ns
NEC	53/695 (7.6%)	7/155 (4.5%)	0.592 (0.275 to 1.277)	ns	21/255 (8.2%)	1.080 (0.665 to 1.753)	ns

USS = ultrasound, BPD = bronchopulmonary dysplasia, ROP = sight threatening retinopathy of prematurity, NEC = necrotising enterocolitis, ns = not statistically significant.

**Table 3 children-09-01706-t003:** Relationships between admission temperature and clinical outcomes.

Outcome	Median (Range) Admission Temperature °C—Outcome Positive	Median (Range) Admission Temperature °C—Outcome Negative	p (Univariate)	p (Multivariate)
Death	36.9 (35.8 to 39.6)	37 (33.6 to 40.4)	ns	
NEC	37 (36 to 40.3)	37 (33.6 to 40.4)	ns	
BPD	37.1 (33.6 to 40.4)	36.9 (35.8 to 40.3)	0.001	ns ^a^
ROP	37.1 (36 to 38.8)	37 (33.6 to 40.4)	0.028	ns ^b^
Major cranial ultrasound abnormality	37.2 (35.8 to 39.8)	37 (33.6 to 40.4)	0.001	0.007 ^c^

^a^ Other items included in multivariate model were birth weight, gestation and gender. ^b^ Other items included in multivariate model were birth weight, gestation and Apgar score at 5 min. ^c^ Other items included in multivariate model were birth weight, gestation and Apgar score at 5 min.

## Data Availability

The data presented in this study are available on request from the corresponding author.

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
