# Peer review of "Very Low Birth Weight Outcomes and Admission Temperature: Does Hyperthermia Matter?"

_children, 2022, doi:10.3390/children9111706_

Round 1
Reviewer 1 Report
Sufficiently powered study with Birthweight < 1500 gms and upto 22 weeks. A breakdown on demographics with percentage of infants < 1000 gms (the highest risk group) would be helpful. Its a retrospective study but relevant as we do see hyperthermia occurring after institution of thermoregulation in DR for VLBW and ELBW babies. Acknowledgement of in-utero environment and chorioamnionitis as causative factors for abnormal cranial US findings is important. It would be interesting to see the subset of babies with no h/o chorioamnionitis with hyperthermia and their subsequent outcome.
Reviewer 2 Report
The present paper reports the results of a very interesting study on the association between birth hyperthermia and clinical outcomes, namely an increased risk of major cranial ultrasound abnormalities.
The discussion is well written and informative. However, I believe that it ends a little too abruptly and that a conclusive paragraph may be needed.
Although the paper is mainly well written, there are still some typos and mistakes that need to be fixed (see for example line 76).
Overall, I consider the paper to be adequate for publication, provided that some minor revisions are applied.
